# Do Embeddings Actually Capture Knowledge Graph Semantics?

Nitisha Jain[1], Jan-Christoph Kalo[2] , Wolf-Tilo Balke[2] , Ralf Krestel[1]

[1] Hasso-Plattner-Institut, University of Potsdam, Germany
[2] Technische Universität Braunschweig, Germany
{nitisha.jain,ralf.krestel}@hpi.de, {kalo,balke}@ifis.cs.tu-bs.de

**Abstract.** Knowledge graph embeddings that generate vector space representations of knowledge graph triples, have gained considerable popularity in past years. Several embedding models have been proposed that achieve state-of-the-art performance for the task of triple completion in knowledge graphs. Relying on the presumed semantic capabilities of the learned embeddings, they have been leveraged for various other tasks such as entity typing, rule mining and conceptual clustering. However, a critical analysis of the utility as well as limitations of these embeddings for semantic representation of the underlying entities and relations has not been performed by previous work.
In this paper, we performed a systematic evaluation of popular knowledge graph embedding models to obtain a better understanding of their semantic capabilities as compared to a non-embedding based approach. Our analysis brings attention to the fact that semantic representation in the knowledge graph embeddings is not universal, but restricted to a small subset of the entities based on dataset characteristics. We provide further insights into the reasons for this behaviour. The results of our experiments indicate that careful analysis of benefits of the embeddings needs to be performed when employing them for semantic tasks.

**Keywords:** knowledge graph embeddings · semantic representation · entity similarity.

## 1 Introduction

Knowledge graphs (KGs) serve as structured repositories of real-world facts in the form of triples comprising of entities and relations e.g. *(head entity, relation, tail entity)*. Popular KGs such as Yago [17], Freebase [4] and DBpedia [2] have been applied to a number of applications including question answering, rule mining and web search. Despite being composed of millions of facts, KGs still suffer from the issue of incompleteness, where entities or facts about entities are missing. A number of solutions have been suggested to cope with KG incompleteness, from statistical methods [27, 22] to the more recent latent embedding based techniques. Following the introduction of the *TransE* embeddings by Bordes et al. in 2013 [5], a flurry of different models have been proposed in the recent years, as summarized by Wang et al. [30].

The fundamental idea behind latent embedding models or knowledge graph embedding models (used interchangeably throughout this paper) is the representation of entities and relations by low-dimensional dense vectors that can capture the interactions within the knowledge graph. Due to their success on the link prediction task towards knowledge graph completion, these models have garnered considerable attention. The intense popularity and frequency of novel ideas towards better KG embedding models has also encouraged the research community to exploit these embeddings for other tasks as well. Since the basic premise of KG embeddings is centered around the semantic relationships between various entities, there is a widespread notion that embeddings must be able to capture the semantics and features of KG entities and relations very well. As such, embeddings have been used for many similarity-based tasks including entity similarity [26] and relation similarity [16], as well as conceptual clustering [9, 10, 29]. Moreover, several previous works have attempted to leverage KG embeddings for performing reasoning with rules [31, 12, 32].

While the results look promising, none of these previous works have performed a detailed analysis of the benefits of the embeddings across different datasets as well as across different entities within a single dataset. In some cases, a measurement of the consistency and scalability of the proposed embedding-based approach for different real-world datasets is largely lacking. The oversight of the limitations of KG embeddings and emphasis on the success for the simpler cases might prove misleading to research community. Our work aims to address this issue by studying the characteristics of the latent vectors obtained from several KG embedding models and quantitatively measuring their ability for semantic representation. With the aid of a systematic evaluation, we report that while embeddings can learn certain semantic features of KG entities on which they are trained, this learning is non-uniform and the quality of semantic representation varies largely across different entities within the dataset.

Our analysis shows that though it seems intuitive to leverage KG embeddings for semantic interpretability (just like word embeddings successfully have been), this is not always the case. The performance of embeddings is, in fact, limited in reality and heavily dependent on the dataset characteristics. We show that even straightforward tasks, such as finding semantically similar entities, do not yield uniformly good results for all entities in the data when relying on the vector representations of these entities. These observations raise doubts about the applicability of KG embeddings not only for semantic reasoning, but also for triple completion and link prediction. Other recent works have also put the efficacy of KG embeddings techniques under scrutiny [24, 25, 1]. These papers evaluate and criticize the KG embedding models primarily in terms of their performance on the link prediction task. In this work, we focus instead on the utility of the KG embeddings for providing semantic interpretations (or rather, the lack thereof). Furthermore, we provide a detailed discussion of the insights from our experimental analysis and identify the factors that determine a good semantic representation of the entities and relations for any given KG as well as the reasons for the shortcomings of current embedding models. We hope our

efforts towards a first comprehensive analysis on semantics in KG embeddings will encourage further investigation into this problem space and assist researchers with a proper inspection of the popular KG embedding models for different semantic tasks. Our datasets and code are publicly available[1].

## 2    Related Work

KG embeddings have been used for a variety of applications over the years. We provide an overview of the related works that follow embeddings-based approach and discuss them in the context of semantics in the embeddings.

**Entity Typing** Finding missing type information for entities in KGs has been a long standing problem. Early techniques usually relied on probabilistic methods for predicting the class membership of entities based on their properties [22]. More recently, KG embeddings have been used together with classification algorithms. As an example, Nickel et al. use RESCAL to predict new type information in a small Yago dataset and show good results on high-level classes such as *persons*, *locations* and *movies* [21]. Moon et al. propose a new embedding technique for performing entity typing [19]. In the example illustration for clustering shown in this paper, it can already be observed that the embedding technique seems to be problematic at distinguishing fine-granular classes such as *author* and *actor*. To a certain degree, their results show that entity typing with KG embeddings is far from being an ideal solution. More recently, an improved embedding technique for entity typing has been proposed [33]. Similar to us, the authors perform an evaluation of embeddings on Freebase and Yago for the entity typing task. While the results already reveal some problems when using entity embeddings for typing, a larger analysis is not performed. In contrast, our work undertakes a detailed analysis of the limits of entity typing when using KG embeddings and shows how classical techniques (e.g. *SDType* [22]) are often superior.

**Entity Clustering** Besides link prediction, entity clustering is another popular application of KG embeddings. In [9], Gad-elrab et al. perform a limited analysis of several clustering algorithms on fine-grained classes. In a related work, the authors leverage rules and embeddings in conjunction to derive explainable clusters from the dataset [10]. However, the results have been shown to work well only for relatively easy relational datasets having well-defined relations between the entities and for small, targeted subsets of Yago. A scalability analysis of these techniques for actual knowledge graphs where their applicability would be most useful is missing. Another related work is presented by Jain et al. [15] where the authors incorporate type information of entities to design better embedding models and demonstrate their results on entity clustering. However, clustering

---

[1]https://github.com/nitishajain/KGESemanticAnalysis

results are illustrated only for limited classes such as persons, organizations and locations without any details on the performance across all classes in the dataset.

Another branch of research concerns with using path-based graph embeddings to perform node classification and clustering tasks [13]. Generally, these techniques aim at creating node (or entity) embeddings using longer paths, instead of relying only on triples like common KG embeddings. However, these techniques are usually evaluated on datasets that do not share the characteristics of knowledge graphs in terms of having fine-grained entity types. Still, as a representative for path-based embeddings, we also evaluate RDF2Vec [23] in this work.

**Other Applications** Besides knowledge graph completion, KG embeddings have been employed in a number of other settings. Similar to previous tasks, it is crucial that KG semantics are captured properly for embeddings to scale well for arbitrary real-world datasets. Embedding approaches have been explored in the context of rule mining on KGs by many previous works with seemingly good results. Existing techniques have either attempted to mine rules directly from the embeddings [31], or use embeddings to support rule mining for confidence computation [12, 32] such that rules of higher quality can be mined. The latter works have not studied or quantified the benefits of embeddings on their work or explored which entities are positively impacted by them.

Furthermore, embeddings are often used to measure the semantic similarity of the entities and relations to perform data integration via entity or relation alignments [16, 7]. An overview of several entity alignment techniques which are based on embeddings is presented in [26]. In our work, embeddings based approaches are compared to classical non-embedding approaches showing no real advantages. This result may already imply that entity semantics is not represented properly in embeddings.

**Criticism of KG embedding models.** For several years, a large variety of knowledge graph embeddings has been developed to perform link prediction to cope with incomplete information in KG. A recent re-evaluation of knowledge graph embedding methods shows several quality problems in the evaluation of KG embedding models as well as the carefully curated benchmark datasets that have been universally used for performance comparison [1]. Akrami et al. demonstrate that existing datasets show several redundancies and cross-product relations. Redundancies in the datasets lead to heavy data leakage thereby making them unrealistically simple in contrast to real-world KG. Furthermore, cross-product relations, connecting all entities to all other entities are frequently used. The authors point out that predictions for these relations is trivial and leads to overestimating the performance of embedding techniques. They show that cleaning the datasets from these defects significantly reduces the link prediction quality of KG embeddings. In another study, the performance gains claimed by newer and more complex models in comparison with the first KG embedding

models has also been questioned [25]. Our work extensively analyzes the problems of current embedding models in terms of their semantic utility, casting doubt on their overall usability in complex real-world KG settings.

## 3 Analysis of the Semantics of Embeddings

In this section, we explain our approach to perform a systematic evaluation of the embeddings for checking their semantic soundness. We also elaborate on the design of our experiments based on popular benchmark datasets.

### 3.1 Categorization of Entities

KG embeddings are trained to capture the structural information of the underlying dataset. Ideally, if latent embeddings were able to embody all the latent features of entities, then entities with similar features would be similar in the vector space as well. That is, entities belonging to a particular *type*, and therefore having similar features would result in similar vectors [29]. Inversely, the embeddings that are close to each other in the vector space would correspond to entities having similar types or features [19]. This implies that it should be possible to identify the entities belonging to a particular type from the KG embeddings. Therefore, in this work we focus on verifying whether the entities can be categorized or assigned to their respective types from their corresponding latent vector representations.

While this is similar to the task of *entity typing* as discussed in Section 2, in this work we chose to follow a comparatively straightforward approach to analyse whether the embeddings in high dimensional space can indeed express the similarities between entities belonging to the same class or concept. We perform a systematic investigation with two distinct sets of *classification* and *clustering* experiments for the entity embeddings in the vector space.

Both these methods are suitable for semantic analysis as they can identify salient features of the embeddings, if any. These can be used to assign the correct class label to the entities in the case of classification, and segregate the entities into separate clusters as per their classes in the case of clustering. If latent embeddings are able to capture the connotations of entities, then this should be reflected in the performance of classification and clustering results obtained by using the embedding vectors as representation. The intentional choice of these techniques is also, in part, to their simplicity, which will enable us to lay the focus on the quality of the embeddings instead of the quality of the evaluation technique itself.

**Classification** With the aid of the supervised approach of classification, we hope to discover the salient semantic features that the latent embeddings are assumed to have learned and use these features to identify the correct class labels for entities. Since an entity can belong to multiple classes in a KG, this entity typing task is a multi-label classification problem where one or many class/type

labels can be assigned to an entity. For our experiments, we employed three different types of classification algorithms which work well for multi-label data. The *Multi Layer Perceptron* (MLP) classifier is a neural-network-based classifier using a simple feed-forward network. We chose the most basic architecture with a single hidden layer with 100 units. As a second classification technique, we chose a *K-Nearest-Neighbour* (KNN) classifier. Lastly, *Random Forest* (RF) classification is used as a decision-tree-based algorithm.

**Clustering**  Being an unsupervised task, clustering is used for identifying the class membership of entities by assigning them to separate clusters, each cluster ideally representing a class. For our experiments, since the ground truth for class labels of entities is known, we are able to measure the quality of clustering by comparing the actual labels with the predicted class labels. Previous works have attempted to identify conceptual clusters in a vector space by applying simple techniques such as *K-Means* to entity embeddings obtained from KG embedding models [10]. We expand our analysis to multiple clustering techniques to weigh the merits and flaws of the techniques and draw conclusions about the characteristics of the underlying embeddings on which clustering is performed. In our experiments, we leverage *Spectral* clustering, *Optics* clustering as well as *Hierarchical Agglomerative* clustering techniques in addition to the simple *K-Means* technique. While hierarchical clustering is particularly suitable for representing the class hierarchy present in most KG ontologies, *Spectral* clustering has shown promising performance for graph based data. *Optics* is a density-based technique that is suited for identifying clusters in spatial data and fits well to our use case.

It is to be noted that our intention for performing clustering in this work is not to discover new concepts but rather to re-discover the existing concepts that the entities are already associated with. Therefore, we provide the required parameter of the number of expected clusters and calculate cluster quality based on ground truth class labels of the entities under consideration.

### 3.2   Datasets

For the experiments, we have chosen the popular benchmark datasets Yago3-10 and FB15K-237. This allows for our results to be put in the correct context with regard to the numerous other related works that have shown good performance on these datasets [6]. Here, we discuss the main characteristics of these datasets and describe the selection of a suitable subset for the clustering and classification experiments.

**Yago3-10**  This dataset was created from the Yago3 knowledge graph [17] by filtering out the entities having less than 10 relations. It consists of a total of 1,079,040 triples with 123,181 entities and 37 relations. Yago is a semantic knowledge base associated with a hierarchical ontology that was derived from *Wordnet* taxonomy [18] combined with Wikipedia categories that are often fine-grained and noisy.

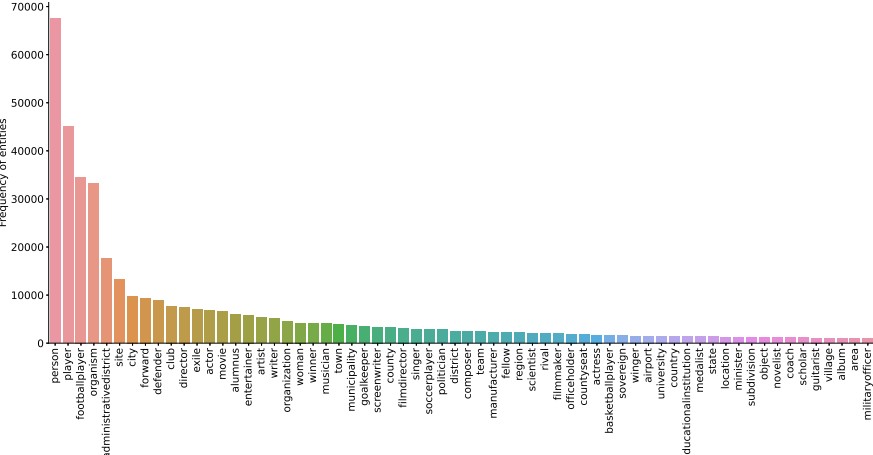

Fig. 1: Yago3-10 class frequency analysis.

In order to explore the differences in semantic representation for entities with varying type granularity, we proceeded to extract entities belonging to classes at different levels of the Yago ontology that resembles a tree-like structure. We limited our analysis to the concepts in Yago that are directly mapped to the *Wordnet* taxonomy to obtain a clean sub-tree of classes that are related to each other. Starting with the main branches of Yago class hierarchy, we chose the classes *person*, *organization*, *body_of_water* and *product*, then progressively explored their sub-trees to design experiments at different levels of the class hierarchy. For this, we manually performed a systematic analysis of the sub-classes of the above four classes and chose the most frequent classes for our experiments. This was a non-trivial task for the Yago3-10 dataset due to the presence of a highly skewed class frequency distribution. As reported previously [11], a large proportion of the entities in this dataset belongs to very few classes, while a long list of classes have very few representative entities. Almost 62% of all the entities belong to the 1% most frequent classes in this dataset. The frequency distribution of the classes (having at least 1000 entities) is graphically represented by Fig. 1 which shows that the class frequency distribution follows Zipf's law.

Due to the constraint of sparse entities in many cases, for each class, a list of sub-classes having entities above a minimum threshold were explored and used for designing the experiments (sub-classes leading to a high skew were omitted to ensure data balance). This was done for three levels starting with the main Yago classes as stated above. Each experiment contains a set of classes that belong to the same level in the ontology. This is important for a fair comparison of the semantic representation of the classes at different granularity levels of the class hierarchy. Table 1 lists all the experiments at different levels along with their classes. For each experiment all the entities belonging to the set of classes

Table 1: Yago3-10 experiments for different levels.

| Experiment | Classes |
| --- | --- |
| Level-1 | person, organization, body_of_water, product |
| Level-2-organization | institution, musical_organization, party, enterprise, non-governmental_organization |
| Level-2-body_of_water | stream, lake, ocean, bay, sea |
| Level-2-person | artist, politician, scientist, officeholder, writer |
| Level-3-person-writer | journalist, poet, novelist, scriptwriter, dramatist, essayist, biographer |
| Level-3-person-artist | painter, sculptor, photographer, illustrator, printmaker |
| Level-3-person-player | hockey_player, soccer_player, ballplayer, volleyball_player, golfer |
| Level-3-person-scientist | social_scientist, biologist, physicist, mathematician, chemist, linguist, psychologist, geologist, computer_scientist, research_worker |

Table 2: FB15K-237 experiments for different levels.

| Experiment | Classes |
| --- | --- |
| Level-1 | person, organization, body_of_water, product |
| Level-2-organization | institution, musical_organization, party, enterprise, non-governmental_organization |
| Level-2-person | artist, politician, scientist, officeholder, writer |
| Level-3-person-writer | journalist, poet, novelist, scriptwriter, dramatist, essayist, biographer |

in the experiment was compiled from the Yago dataset, then the corresponding embeddings for these entities was extracted from pre-trained KG embeddings models to serve as data for the clustering and classification experiments.

**FB15K-237** This second dataset is a subset of the Freebase knowledge graph, frequently used by knowledge graph embedding models. FB15K-237 [27] comprises 272,115 triples with 14,541 entities and 237 relations. It was derived from the FB15k [5] dataset by filtering out redundant and inverse relations. With regard to the domains, it mainly pertains to *persons*, *organizations* and *products* and we aimed to design our experiments with a similar structure. We performed the mapping of Freebase entities to Yago through existing *sameAs* links and chose classes and sub-classes by following the Wordnet taxonomy. The experiments were designed in the same way as described above for the Yago dataset for allowing direct comparisons. The Freebase dataset is significantly smaller than

the Yago dataset, such that the number of entities reduces dramatically when considering the classes at level-3. Therefore, we had to limit ourselves to fewer experiments as listed in Table 2.

### 3.3 Knowledge Graph Embeddings

For all the experiments, we obtain the pre-trained embeddings models for the benchmark datasets from the LibKGE library [6] since extensive hyper parameter tuning has already been performed. We used five different embedding techniques that are widely popular : TransE[5], RESCAL[20], Complex[28], DistMult[31] and ConvE[8]. Since for Yago3-10 only the Complex embeddings were available, we trained the remaining embeddings ourselves by adapting the parameters that were used for the Freebase dataset[2]. Another popular branch of embedding approaches is based on paths in a knowledge graph, usually showing good results in entity typing tasks as discussed in Section 2 [23]. RDF2Vec was trained using paths created by a random walker algorithm which created paths of length 4. Then the model was trained for 50 iterations using pyRDF2Vec library[3].

## 4 Experiments

In this section we present the results of our experiments for clustering and classification on Yago3-10 and FB15k-237 datasets. Additionally, we draw comparisons with a traditional statistical approach.

### 4.1 Non-Embedding Baseline

To ensure that the results are not driven solely by the performance of clustering and classification algorithms, we found it important to include a baseline that is unrelated to the embeddings. For this, we leveraged the *SDType* approach as introduced by Paulheim et al. in 2013 [22]. This is a heuristics based technique that simply uses the links between the entities to infer their type. Based on the incoming and outgoing relations associated with a particular entity, the average probability of each type for an entity is calculated. Purely relying on the statistical distributions of the entity links, this method is robust to noisy facts in the dataset and agnostic to existing type information. We rely on this approach to stipulate whether any semantic features are present in the underlying data that can help with the deduction of type information for the entities. If the statistical approach can already leverage the semantic features in data to identify the types for entities, this indicates that unsatisfactory scores for classification or clustering on embeddings must be due to the failure of embedding models to capture these semantic features during training. We report the performance of *SDtype* for our experiments along with the classification results in terms of the best F1 measure obtained (P-R curves are available on github link).

---

[2]The training parameters and performance scores are available on github link.
[3]https://github.com/IBCNServices/pyRDF2Vec

## 4.2   Evaluation Metrics

Similar to previous works [10], we measured the Adjusted Rand Index (ARI), Normalized Mutual Information (NMI) and the V-measure to estimate the quality of the clusters. With the true and predicted labels as input, ARI measures the similarity of the assignments with values between -1 and 1 (0 stands for random assignment, 1 is the perfect score). NMI measures the agreement of the assignments and V-measure is the harmonic mean of homogeneity and completeness of the clusters. For both, the values lie between 0 and 1, with 1 being a prefect score. For the evaluation of classification experiments, an 80-20 ratio was used to split the dataset (consisting of entity embeddings and class labels) into train and test set. Since the task is a multi-label classification, the weighted average of F1 measures per class (in %) in the test set was used as an evaluation measure.

## 4.3   Classification Results

Fig. 2 shows the weighted F1 measures for Yago3-10 dataset across all the embedding models (color coded) as well the different classifiers (pattern coded). It can be seen from this figure that all the classifiers perform very well for level-1 experiment (refer to Table 1), where the considered classes are coarse-grained and distinct from one another. However, the performance starts degrading once experiments at level-2 are considered and becomes worse for level-3, where the F1 measure drops below 20 for sub-classes of the *scientist* class. This is due to the fact that classes are finer-grained for these experiments, where they all have a common parent class and share certain common features. For instance, different types of *persons*, and further, different types of *artists*, *scientists* etc. would all share common properties of the *person* class (discussed in detail in Section 5). Even though the considered classes are conceptually distinct from one another, the classification algorithms find it hard to perform label matching correctly based on embeddings. This behaviour is uniform across all clustering algorithms and all embedding models, with no setting performing particularly better or worse. Though fine-grained entity typing is indeed a hard problem, our experiments are designed only for the top three levels of classes. It is indicated by these results that embeddings simply do not possess the necessary semantic features such that classification could identify correct entity types beyond the highly coarse-grained classes.

Similar trends are also seen for the FB15k-237 dataset (Fig. 3) where classification performs very well for the level-1 experiment, but gets worse progressively for level-2 and level-3. A few exceptions in this trend are noticed when the dataset is highly skewed towards entities of a particular class, such as *players* in case of Yago and *artists* in case of Freebase. In this case, the performance is improved to some degree as compared to other experiments at the same level. The performance of Freebase is generally better than Yago due to the presence of more relations in the dataset. Overall, the drop in classifier performance with increasing levels indicates a lack of sufficient semantic representation in embeddings for fine-grained entities for both the datasets.

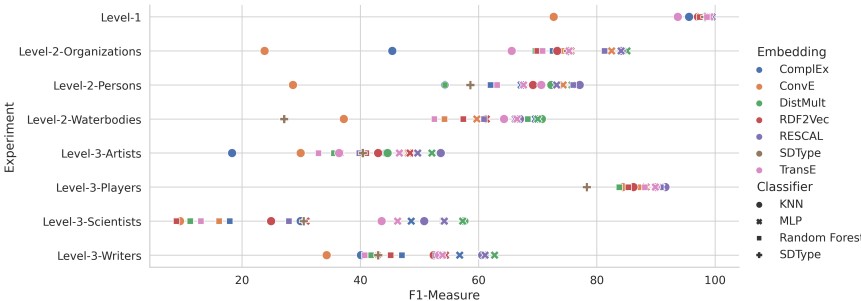

Fig. 2: F1 measure for Yago3-10 classification experiments.

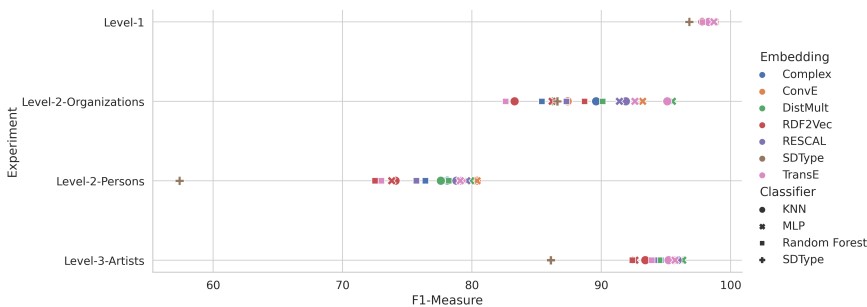

Fig. 3: F1 measure for FB15K-237 classification experiments.

To compare and contrast the performance of the *SDType* baseline approach, the F1 measures for *SDType* are also shown in Fig. 2 and Fig. 3 (coded with a different color and symbol). Significantly, it can be seen that *SDType* is able to achieve quite competitive results as compared to the embeddings, notably for the level-3 classes. This provides strong evidence for the shortcomings of embeddings for representing fine-grained classes for which even simple statistical approach can already give comparable results.

### 4.4 Clustering Results

The results for the clustering experiments are reported in terms of the NMI scores and shown in Fig. 4 for the Yago3-10 dataset and Fig. 5 for the FB15k-237 dataset. Overall, clustering performs worse than classification, which raises doubts over the expected spatial closeness of similar entities in the vector space. Further, the clustering results also demonstrate a similar pattern to the classification results. The NMI scores are relatively better for level-1 classes but get progressively worse for lower levels[4]. All embedding techniques fair similarly, thus conveying that it is difficult to identify or re-discover even the existing entity types or classes from any of the embeddings with the help of clustering,

---

[4]ARI and V-measure show similar trend, full results are available on github link.

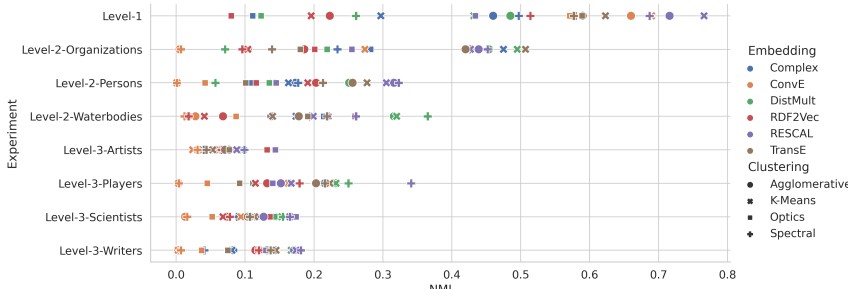

Fig. 4: NMI measure for YAGO3-10 clustering experiments.

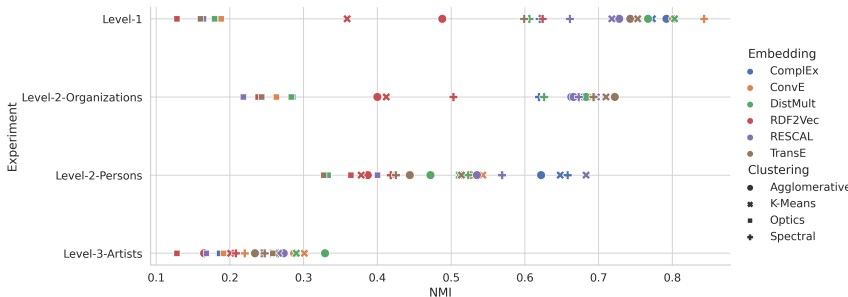

Fig. 5: NMI measure for FB15K-237 clustering experiments.

except for very high-level classes. Considering the different algorithms, *Optics* shows worse clustering scores in many cases. Since *Optics* is a density-based clustering technique, the low quality of clusters again point towards the lack of proper conceptual representation in the embeddings in vector space.

## 5  Discussion

From the experimental results on both supervised and unsupervised tasks, it is clear that KG embeddings are unable to capture the latent features that would be sufficient for a good semantic representation for all entities of a KG. While entities belonging to a small set of high-level *easy* classes are relatively well-represented, the same does not hold true for most of the entities corresponding to other important classes in the dataset. We investigated further to understand the plausible reasons for this shortcoming and discuss our findings here.

Looking beyond the flaws in the training and evaluation process of the KG embedding models (that has been the focus of previous works as discussed in Section 2), we studied the characteristics of the underlying KG datasets on which the various embeddings are trained. Knowledge graphs such as Yago and Freebase are comprised of real world entities that frequently belong to more than one semantic type or class e.g. an artist can also be a politician in real

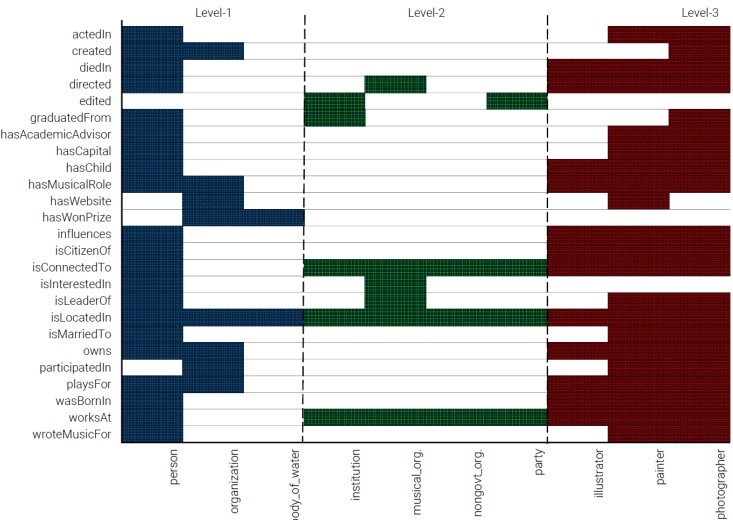

Fig. 6: Representation of outgoing relations at different levels in Yago.

life. Since such entities would reflect the characteristics of multiple classes, they are associated with a number of different relations that are neither unique nor indicative of any single class in particular.

To explore this further, we performed an analysis of the relations associated with the different classes that were used in our experiments for the Yago3-10 dataset. For each class, the incoming and outgoing relations associated with all the entities of the class were separately identified. Thereafter, the classes were compared to each other in terms of their relations within the same experiment as well as across experiments at different levels (as listed in Table 1). Fig. 6 shows a comparison for classes at different levels based on their outgoing relations for a few representative experiments. Here, a slot is shaded depending on the premise that the relation was found for a minimum number of entities of the class. The figure demonstrates that the classes at level-1 have different sets of relations associated with them, i.e. there are few overlapping relations. This is less so for level-2 classes where several relations are found to be common. Finally, at level-3 there are hardly any unique relations that could distinguish one class from another and the relations overlap is quite substantial.

These results stem directly from the characteristics of real-world data where, for instance, all persons have similar properties (e.g. *wasBornIn*, *isCitizenOf*) regardless of their profession. In Yago3-10, any specific relations that could have uniquely identified, e.g. an *artist* from a *politician* seem to be either missing or very sparse. This directly affects the embeddings since they are trained to learn the associations between the different entities of a KG (a heuristics based approach like *SDType* can exploit sparse links much better). The presence of overlapping relations among entities belonging to different semantic types hin-

ders their ability to encapsulate *type-specific* features. In this case, an embedding model can only hope to learn from other entities that are found in the triples of the entities of a particular class, and find patterns and features from those entities. However, recent work has shown that relations in knowledge graphs can be ambiguous in the way they connect different entities [14]. This means that various types of entities might be connected to a particular entity by the same relation. Such generic and noisy links make it even harder for embedding models to derive type-specific features about the entities, thus limiting their capability to learn similar entities or identify any common traits for all entities belonging to the same class. It is worthwhile to note that some classes such as *musical_instrument* and *tv_program* in Freebase have been shown to cluster well in the vector space [19]. A closer inspection reveals that these classes have very few and unique incoming relations such that the embeddings would be able to learn their features well. However, classes with unique representative properties are not very common in real-world datasets.

The key insight from our detailed analysis in this work is that while KG embeddings are assumed to be representing the semantics for entities and relations, in reality their semantic soundness is severely restricted and highly dependent on the datasets on which they are trained. Experimental results have clearly shown that several prominent embedding models often record worse semantic capability for a majority of the entities in real-world datasets as compared to a simple heuristics based approach that can derive the semantics directly from KG triples without any additional information. These findings indicate that a thorough inspection of the advantages and weaknesses of KG embeddings is necessary when employing them for semantic tasks. While the semantic web community is focused on novel architectures for training the KG embeddings models, a careful eye on the generalizability of these models in terms of their semantic representation also deserves more attention. We hope this work will guide further research in this direction. Recent efforts towards the explainability in KG embedding models [3, 10] could be the first steps towards understanding these models that could benefit all semantic tasks that leverage them.

## 6 Conclusion

In this paper, we performed a comprehensive analysis of the popular knowledge graph embedding models in terms of their semantic utility. The results from our classification and clustering experiments on top of these embeddings brings attention to the weaknesses in semantic representation of embeddings. We have shown that embeddings fare poorly in terms of identifying the concepts or classes for a majority of the entities in the underlying knowledge graph and simple statistical approaches can compete very well with them. We also presented a detailed analysis of the reasons for limited semantic understanding of the embeddings relating to sparse and noisy links in real-world datasets. We hope the results from this work would serve as a precautionary tale and help the research community become cognizant of the realistic semantic benefits of knowledge graph

embeddings, such that they can make prudent decisions when applying these embeddings to new problem statements and semantic tasks. We plan to extend this analysis to include further and more recent embedding techniques.

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
