# OpenReview forum: "Do Embeddings Actually Capture Knowledge Graph Semantics?"
_eswc-conferences.org/ESWC/2021/Conference/Research_Track — ESWC 2021 Research_

### Official Review · AnonReviewer2 · 2021-01-11
**The paper presents interesting criticism of widespread Knowledge Graph embedding models. The authors designed several experiments to show that KG embedding models did not capture semantics in the case of realistic datasets. While the paper is clearly written and I think it presents original ideas with a widespread extent, it is unclear how it is ensured that employed datasets are realistic and the authors did not show that other datasets are unrealistic and simple.**

**Rating:** 1
**Confidence:** 4
**Impact:** 4
**Design And Technical Quality:** 3

**Review:**

I acknowledge that I have read the rebuttal. I would like to thank you to the authors for their answers. I modified the review accordingly.

==

Nowadays embeddings is a hot topic for many fields with regard to a representation. Knowledge Graph Embedding Models are widely used in Semantic Web. Systems based on embeddings usually work well but the question is how it is actually with semantics of represented entities or relations. This interesting question is the main topic of the paper. The authors claim that the semantics of KG embedding models is not adequately questioned within the literature. The common assumption is that KG embeddings are well enough to capture the semantics of KG entites and relations. As a result, embedding models are widely used not only for entity typing and link prediction but also for entity clustering and rule mining. However, the semantics of entities captured by embeddings remain questionable.

On the one hand there can be a lack of semantics in KG embeddings, on the other hand KG embeddings can still work very well in applications such as link prediction or entity typing but this paper claim that embeddings' great performance is achieved at the expense of unrealistic simple datasets.

Therefore, the authors propose the experiments over realistic datasets (YAGO and FreeBase derivatives) dealing with entity classification and entities clustering. The creation of datasets are thoroughly explained but yet these are not available online (as promised). The most frequent classes are selected for experiments. From the current paper it seemed to me that the authors criticized the other datasets as being unrealistic but as explained in the rebuttal the criticism is merely targeted at semantics of embedding techniques. This should be explained better in the text.

Besides datasets several embedding techniques were employed in the experiment along with various classification/clustering algorithms and one non-embedding approach (SDType). The obtained results show that embeddings work worse for finer-grained classes in both applications due to sparse and noisy information on which representation has been learned. While this can be clearly seen from the experiments on YAGO dataset, this is not so straighforward with regard to FreeBase dataset where embeddings work better on level 3 than on level 2. Please provide an explanation of this in the paper.

The authors also mentioned that datasets commonly used for evaluation of embedding models are unrealistic and simple. Is there any evidence of this characteristics? Did authors perform similar detail analysis as presented in Section 5 also on other (unrealistic) datasets?

The paper is clearly written and since I think it presents original ideas with a widespread extent, I am curious about the answers on my abovementioned questions.

Minor comments
* p. 3, "Besides link prediction" -> Besides entity typing?
* p. 10, "refer to Table. 1"
* p. 11, It is not clear why Level 1 is above Level 2. What is the rule for order of experiments in the figures?
* p. 12, Figure 3 and Figure 4 should contain clustering algorithms instead of classification algorithms

**Anonymity:**

Yes, I would like my review to remain anonymous.

**Reuse And Availability:**

3: Medium

**Strong Points:**

- original criticism of widespread representation models
- thoroughly discussed experiment
- a potential widespread impact

**Subreviewer:**

I submitted this review.

**Weak Points:**

- Authors did not show that other datasets are unrealistic and simple
- the datasets and code are not yet available but it is written that "The datasets and code will be updated here as well."

---

> ### Author Rebuttal · Authors · 2021-01-30
>
> Dear reviewer, we would like to thank you for the detailed review and analysis of our paper. We will address the minor comments in the final version of the paper. Please find our response to other comments and questions as follows:
>
>  * *how one can be sure that these data are realistic or more realistic than usually used datasets which the authors criticized?*
>     We would like to clarify that this work is not aimed towards analyzing the datasets that are used for creating the embeddings but rather towards analyzing the semantic capabilities of the embeddings trained on widely used datasets. We have chosen two representative real-world datasets to illustrate and analyze this issue, but this can be extended to other datasets as well.
>  We have mentioned the simplicity or unrealistic nature of the datasets in the context of discussing a previously published related work (*Akrami et al. Realistic re-evaluation of knowledge graph completion methods: An experimental study*) which looks deeper into this issue. However, our analysis is quite different in terms of the goals, in that we instead look into the semantic shortcomings of the embedding techniques irrespective of the datasets that they are trained on. We do not claim that embeddings' great performance is achieved at the expense of unrealistic simple datasets, we are in fact demonstrating that the embeddings can perform well for the *easy* classes of any dataset, but the performance does not scale to all the classes in the considered dataset. We will be sure to make this explicitly clear while discussing previous work to avoid confusion.
>
> * *Another related question is whether authors had also in mind the consistency and scalability issues/measurements mentioned on page 2 as often lacking from datasets?*
>  Regarding the consistency and scalability issues/measurements mentioned on page 2, we want to convey that methods that use embeddings for semantic tasks have not performed such an analysis in general, if such an analysis was indeed done by previous work, then the limitations of the embeddings could probably have been more visible. Since this is missing from previous work, therefore we have focused on this issue and performed dedicated experiments to illustrate the same.
>
> * *..this is not so straightforward with regard to FreeBase dataset where embeddings work better on level 3 than on level 2. Is there any explanation about this inconsistency?*
> One possible explanation for this outlier could be the availability of more triples for some classes at lower levels in comparison to classes at higher levels. However, the overall results indicate towards worsening performance with finer-grained classes in general.
>
> * *the datasets and code are not yet available*
> We are in the process of making the entire dataset and code available on github, this is already partially done and we will finish it within the next few days.

---

### Official Review · AnonReviewer5 · 2021-01-13
**Review of analysis of embedding approaches**

**Rating:** 2
**Confidence:** 4
**Impact:** 3
**Design And Technical Quality:** 4

**Review:**

The paper analysis knowledge graph embeddings, based on two existing knowledge bases: Yago3-10 and FB15k-237. The hypothesis of the authors is that quality of embeddings is only sufficient for a small subset of entities. They validate this hypothesis by applying various embedding techniques to the data sets and by analysing the results against a non-embedding baseline and various classification and clustering algorithms. Their findings show that different types of entities can only be identified on a high level, e..g "person" versus "country", whereas on a lower level, it is hard to differentiate the types, e.g. "politician" versus "artist".

Pros: The paper is

* Well written in terms of quality and clarity
* Provides a clear approach for evaluating knowledge graph embeddings

Cons: The paper

* has findings that are not a surprise, e.g. that properties of different occupations ("politician" versus "artist") are hard to differentiate
* misses approaches of working with sub sets of knowledge graphs, see e.g. https://arxiv.org/abs/2009.07659
* misses the ability of certain approaches like RDF2VEC, to parametrize the embedding approach. For RDF2VEC, the authors use a random walker, but the underlying library allows to define walking strategies, see https://pyrdf2vec.readthedocs.io/en/latest/readme.html#define-walking-strategies-with-their-sampling-strategy
* focuses on the shortcomings of knowledge graph embeddings, without proposing a way to improve knowledge graph embedding approaches

**Anonymity:**

Yes, I would like my review to remain anonymous.

**Reuse And Availability:**

4: High

**Strong Points:**


The strength of the paper is that

* it uses various methods to validate the quality of knowledge graph embeddings
* the validation is based on large existing knowlede bases
* the validation takes into account several approaches for generating knowledge graph embeddings.

**Subreviewer:**

I submitted this review.

**Weak Points:**

The weak points of the paper are that

* it has findings that are not a surprise: that is hard to differentiate more fine-grained types, compared to corse-grained types
* it evalutes its approach only with two knowledge bases which are rarely used in real-life applications
* it does not take into account ways to parametrize knowlede base embedding approaches.

---

> ### Author Rebuttal · Authors · 2021-01-30
>
> Dear reviewer, thank you very much for your optimistic review of our paper.
> We would like to mention that the choice of the datasets as well as the KG embedding techniques was motivated by their frequent use in several research works concerned with this topic. As this is a negative results paper, we wanted to make sure that we are relatable to other works and perform our analysis on the popular datasets and techniques. We did not propose any new embedding approach but rather performed a critical analysis of existing embedding techniques and reported negative results as our contribution.
> Thank you for providing pointers for different approaches and their parameterization, we will include these as part of an extended study in future work, such that the larger research community can be made aware of the semantic limitations of the embeddings.

---

### Official Review · AnonReviewer4 · 2021-01-13
**Results are welcome, but the way of evaluating embeddings is not well justified**

**Rating:** 1
**Confidence:** 3
**Impact:** 3
**Design And Technical Quality:** 3

**Review:**

I believe the authors are sutdying an important problem: models and use cases for graph embeddings are skyrocketing, but we are laggin behind in evaluating whether these tools are such a good fit for Knowledge graphs. In this sense, I agree with the motivation of the paper, and I share the criticism exposed by the authors.

I also believe that the way of testing embeddings proposed in this article is reasonable, and I think the hypotesis resting behind the plan is the following: if the graph embeddings actually do capture semantic knowledge, then one should expect good results when we feed the embedded graph to a) a classification task trying to guess the class labels of a given entity, and b) a clustering task that tries to group all entities belonging to the same class label.

Results show that embeddings work when one tries to classify/cluster entities into the highest level of a class hierarchy, but start failing once we drill down on this hierarchy.

One critique I have to this paper is that, as a baseline, I would expect any automatic classifier/clustering algorithm to behave much worse when we go down in the hierarchy. Even for humans, it is easier to distinguish a rock from a personthan, say, and actor from a musician.

And another critique is that the hypothesis I mentioned in the first paragraph is taken as a given: there is no discussion about it.

As such, this paper would be much stronger with
- a deeper discussion about why classification / clustering tasks are a good way to assess embeddings, and
- a more refined baseline regarding the natural drop in accuracy when we move to lower levels of the class hierarchies in the dataset (and therefore to more complex classification/clustering examples). What would one expect to see in order to say with confidence that "embeddings DO capture the semantics of lower-level classes"?
- another aspect about the discussion in page 14, is that I am not able to distinguish whether the following is a problem of the embedding approach, or solely of the algorithms considered in this evaluation: "Such generic and noisy links make it even harder for embeddingmodels to derive type-specific features about the entities, thus limiting their capability  to  learn  similar  entities  or  identify  any  common  traits  for  all  entitiesbelonging to the same class."


**Anonymity:**

Yes, I would like my review to remain anonymous.

**Reuse And Availability:**

3: Medium

**Strong Points:**

- the problem studied is important
- the critique of kg embeddings is motivated in previous work, and discussed in the paper in a satisfactory way
- the set of experiments are a good way to test the hipothesis in the paper

**Subreviewer:**

I submitted this review.

**Weak Points:**

- the hypothesis in the paper (that the quality of embeddings can be witnessed by classification/clustering) is not properly discussed
- the baselines for the experiments is complicated to follow. What would one expect to see in order to answer whether embeddings capture semantics or not?

---

> ### Author Rebuttal · Authors · 2021-01-30
>
> Dear reviewer, thank you very much for your interesting comments.  Please find our response as follows:
>
> * *A deeper discussion about why classification/clustering tasks are a good way to assess embeddings..*
> This is a very important point and we understand that it is crucial to discuss and explain the motivation for the design of our experiments. We have indeed explicitly addressed this point in the paper in *Section 3.1*. In particular, we would like to refer to the 3rd paragraph of this section where we discuss the classification and clustering tasks and motivate why they are a good choice for evaluating the semantic representation of the embeddings. We will rephrase and make this point clearer in the final version.
>
> * *What would one expect to see in order to say with confidence that "embeddings DO capture the semantics of lower-level classes"?*
> A very relevant question indeed. While we do not have an absolute answer, it can be speculated that the embeddings should show similar performance for all classes having similar representation (in terms of no of KG facts) regardless of the class hierarchy, if they were indeed capturing the semantics correctly. Since a relatively straightforward statistical approach is seen to be achievable comparable results to embeddings, the utility and advantages of embeddings for semantic tasks become questionable. While defining an ideal baseline performance is hard, the motivation of this work is to bring attention to the limiting behavior of the embedding techniques, which is often overlooked while employing them for semantic tasks. We will clearly highlight this point in the paper in the hope of encouraging further discussion towards possible solutions.
>
> * *Such generic and noisy links make it even harder for embedding models to derive type-specific features about the entities, thus limiting their capability to learn similar entities or identify any common traits for all entities belonging to the same class.*
> We would like to clarify that the underlying KG datasets having noisy and generic links is being identified as one of the reasons why the embeddings are unable to capture the semantics of the classes. Therefore, it is the *approach* of creating the embeddings from the KG datasets, regardless of the algorithms, whose semantic capability is being questioned in our work. We choose multiple algorithms to illustrate this issue, but the fundamental reasons for this are agnostic to the algorithmic techniques considered in this work and should generalize to any other algorithm as well (we will analyse this further in future work).

---

### Official Review · AnonReviewer1 · 2021-01-18
**Interesting analysis but miss comparison with more recent work such as KG-BERT**

**Rating:** 1
**Confidence:** 3
**Impact:** 4
**Design And Technical Quality:** 3

**Review:**

This paper performs an analysis of knowledge graph embedding approaches for their ability for representation learning and their ability to capture the semantics of the entities and relations in knowledge graphs. The semantic capabilities of the learned knowledge graphs embeddings are evaluated using the performance of classification to label entities to their corresponding types and clustering similar entities together. The topic of the paper is very relevant to the ESWC community and it’s timely given the increasing popularity of using knowledge graph embedding for many downstream tasks. The paper is well-written and easy to follow. The paper would fit well as a negative result.

Comments:
- I wonder if the subset of data selection can be done in a more systematic way. Currently, it only considers the granularity of the class such that the level in the type hierarchy. I wonder if the classes could have been sampled in other dimensions such as number of unique relations, number of incoming/outgoing links, etc. so that performance on each category can provide further insights of high/low performances and different capabilities of the embeddings.
- The other point that would be interesting to further analyse is that if there is an upper-bound for performance such that entities actually contain enough information to assign them types only using structural information. I think SDTypes baselines provide some clues on this but it is not clear in some other settings.
- Continuing on the previous point, it would be very interesting if the more recent knowledge graph embedding models such as KG-BERT are included in this comparison. https://ojs.aaai.org/index.php/AAAI/article/view/5681
- While I appreciate the visualization and understand the motivation to save space in Fig 2. / 3. / 4. / 5. , they are not easy to follow. One alternative is a table with values but I can imagine it would be bloated with too many values. I suggest trying some other ways to make those figures easy to read and comprehend.
- Table 1. Level-2-person is missing `player`, isn’t it?

**Anonymity:**

Yes, I would like my review to remain anonymous.

**Reuse And Availability:**

3: Medium

**Strong Points:**

- The paper discusses an interesting and relevant topic to the community
- The paper provides a set of quantitative experiments and attempts to provide qualitative explanations to understand the results.


**Subreviewer:**

I submitted this review.

**Weak Points:**

- The paper doesn’t include comparisons with more recent approaches such as KG-BERT.
- There is no baseline for comparison for clustering experiments.
- Results don’t seem to provide conclusive evidence and maybe further analysis is needed.

---

> ### Author Rebuttal · Authors · 2021-01-30
>
> Dear reviewer, thank you for your insightful comments. Please find our responses on the points you have raised as follows:
>
> * *I wonder if the subset of data selection can be done in a more systematic way..*
>     This is a very good point and the topic of much discussion and deliberation for this work. We decided to go with the granularity of the classes in the type hierarchy for dataset design since it is already correlated with other features such as relation cardinality and uniqueness. We have presented this analysis in detail in Section 5, where we explicitly illustrated in Fig. 6 that the classes at a lower level have fewer unique relations. We also touched upon (though briefly due to space constraints) regarding the good clusters observed for entities such as *tv_program* that only have incoming relations that are few and unique.
>      During initial analysis, we had observed that relation uniqueness plays a greater role in affecting the embeddings as compared to the cardinality of the relations, e.g. different types of institutions have a few relations associated with them, while different types of persons have many incoming/outgoing relations, but in both cases, embeddings fail to distinguish between the subclasses of institutions or persons. Therefore, we focused on the level of the classes to design the dataset.  We hope this makes our motivations a bit clearer. We will make sure to address this point and make it explicitly clear in the final version.
>
> * *..if there is an upper-bound for performance such that entities actually contain enough information to assign them types only using structural information..*
> This is again an excellent point, but one which is hard to address. In order to estimate the achievable semantic information from the KG triples for the task of entity typing, we explored the statistical approach of SDType for comparison. However, as there is no approach available which has solved the entity typing task entirely as per our knowledge, there is no clear way to decide an upper-bound for the semantic information.
>     The focus of our work is to demonstrate that embeddings perform no better than previously available statistical techniques and question the suitability or advantage of using embeddings for semantic tasks. While the idea of upper-bound for semantic capability is appealing and certainly would be interesting to address, it goes beyond the scope of our work. We will include this idea in the last section of the paper and hope to encourage further discussions in this direction. We would also like to mention that we were unable to find a similar suitable baseline for the clustering experiments and would appreciate any pointers in that regard.
>
> * *it would be very interesting if the more recent knowledge graph embedding models such as KG-BERT are included in this comparison*
> We agree that it would be useful to extend this study for several other recent KG embedding techniques, including KG-BERT. We plan to do this as part of future work.
>
> * *Visualizations*
>  While we agree that the figures pack a lot of information in terms of different embedding and classification/clustering techniques, we have chosen to illustrate the results this way so as to enable a joint comparison between the different techniques in all settings. Since the different settings do not exhibit any specific or significant differences, the main message from the plots is that the performance varies with the level of class hierarchy and we believe that this is clearly conveyed in the figures. We have mentioned other noteworthy findings in the text and we will ensure that nothing is left unaddressed in the final version.
>     We would also like to mention that the detailed and complete results for deeper analysis have been made available in the supplementary material at https://github.com/nitishajain/KGESemanticAnalysis. We will further add figures addressing each clustering/classification technique individually for better understanding of the interested reader.
>
>  * *`Table 1. Level-2-person is missing player..*
>     Noted, we will fix in final version.

---

> > ### Comment · AnonReviewer1 · 2021-02-11
> > **Thank you for the clarifications.**
> >
> > I would like to thank the authors for the clarifications on the points raised.

---

### Official Review · AnonReviewer3 · 2021-01-20
**Analysis of KGE semantics on different tasks. However, a lack of other KGE approaches takes place**

**Confidence:** 5
**Impact:** 3
**Design And Technical Quality:** 3

**Review:**

The authors analyzed the semantics of knowledge graph embeddings on different tasks. The paper is well-written and easy to follow. Additionally, the investigation performed in the paper is totally relevant and the crucial importance for all communities. However, to support such claims and findings, the authors should have analyzed more KGE techniques, and not only five approaches. I did enjoy reading the paper and the study, but to have it published, I suggest the authors evaluating other approaches (see table 3 in [1] for a good list of KGE works). Additionally, the authors could have performed other tasks for analyzing the semantics of KGE (see [2] for some inspiration)

[1] Demir, C., & Ngomo, A. C. N. (2020). Convolutional Complex Knowledge Graph Embeddings. arXiv preprint arXiv:2008.03130.
[2] Petroni, F., Rocktäschel, T., Riedel, S., Lewis, P., Bakhtin, A., Wu, Y., & Miller, A. (2019, November). Language Models as Knowledge Bases?. In Proceedings of the 2019 Conference on Empirical Methods in Natural Language Processing and the 9th International Joint Conference on Natural Language Processing (EMNLP-IJCNLP) (pp. 2463-2473).

**Anonymity:**

No, I would like my review to be deanonymized.

**Rating:**

-2: Reject

**Reuse And Availability:**

3: Medium

**Strong Points:**

- interesting and promising study of semantics in KGE

**Subreviewer:**

I submitted this review.

**Weak Points:**

- lack of KGE approaches to support the findings
- the paper requires more tasks

---

> ### Author Rebuttal · Authors · 2021-01-30
>
>
> Dear reviewer, we would like to thank you for your feedback on the paper.
> For our analysis, we attempted to select the representative KGE models that are popularly used as baselines by several papers that are concerned with extending or criticizing the embedding techniques.
> We appreciate and agree with your comments on extending the evaluation further to support the findings and we plan to incorporate the same in a follow up longer version of the study in the near future.

---

### Decision · Program_Chairs · 2021-02-23

**Decision:**

Accept

**Comment:**

All reviewers appreciated that this paper addresses a very important and timely problem, in the rapidly evolving field of KG embedding.  The proposed experimental approach and  the clarity/quality of the writing were additional strong points of the paper.
However, here was quite a wide-spread concern that the scope of the work is not sufficiently broad to provide meaningful findings and in particular that it is not sufficiently timely by not covering more recent KG embedding methods.

Given the importance and timeliness of the problem, it's relevance to the community, the potential of this work to fuel discussion at the event as well as the difficulty to provide timely analysis in this rapidly evolving field, the PC felt that the findings of this paper are more valuable to the community now than in a few months (in case the paper was rejected and needed re-submission). Therefore, the paper was accepted.

Besides addressing reviewer comments for the CR, the authors are asked to also include some considerations about the rationale for selecting the KG embedding methods that they compared, in order to better justify the scope of the work.